# Impact of Unlisted Small and Medium-Sized Enterprises' Business Strategies on Future Performance and Growth Sustainability

**Won Park [1], Chang Soo Sung [2] and Chung Gyu Byun [3,\*]**

[1]  Department of Business Management, Uiduk University, Gyeongju 38004, Korea
[2]  Department of Technology Entrepreneurship, Dongguk University, Seoul 100-272, Korea
[3]  Department of Business Administration, Gyeongsang National University, Jinju 52828, Korea
\*  Correspondence: cgbyun@gnu.ac.kr; Tel.: +82-055-772-1512

**Abstract:** This study aims to identify, from among business strategies presented in Miles and Snow (1978, 2003), those that increase the future performance or growth sustainability of unlisted small and medium-sized enterprises (SMEs) in Korea. (This study applied the scope of SMEs based on Article 2 of the Enforcement Decree of the Tax Exemption Restriction Act in Korea. The Korean tax law sets the scope of SMEs based on independence criteria set by the scale of sales, assets, and affiliation, and subordination. For example, the size standard of a manufacturing industry can be regarded as a small business if the average sales amount is less than 150 billion won or the total amount of assets is less than 500 billion won for three years.) In addition, it analyzes measured variables of business strategy for factors influencing unlisted SMEs' future performance and growth potential. The objective is to determine a business strategy for unlisted SMEs, which are small, financially challenged, and have shorter lifespans and faster growth rates. The results highlight that investment in research and development (R&D) for new products influences both future performance and growth sustainability. R&D is an important intangible resource of the firm, which increases future risk due to high future uncertainty, but it is also an important factor to increase future performance or value based on resource-based theory. In the accounting field, research and development expenditure also provided evidence of future excess return or performance. This study is meaningful because it identifies the role of research and development in increasing future growth sustainability in SMEs, thus emphasizing change and innovation.

**Keywords:** business strategies; future performance; growth sustainability; prospector; defender

---

## 1. Introduction

   A firm's business strategy is an important decision-making factor in terms of objectives, marketing, production, finance, recruitment, human resources, and investment activities, thus influencing its overall performance [1,2]. Therefore, it is imperative for firms to choose appropriate business strategies on the basis of their characteristics and economic conditions [3] since failure to do so could hinder their ability to generate excess profit and as a result, they become difficult to maintain. Unlisted small and medium-sized enterprises (SMEs), in particular, struggle to recruit external investors and maintain net cash flow if they are unable to report consistent performance. To address these issues, SMEs must engage in strategies that can support their long-term survival.

   Porter [4] distinguishes between price advantage and differentiation strategies using a generic approach. Alternatively, Miles and Snow [5,6] suggest that the strategy choosen by the firm in a continuous line can distinguish between prospector and defender strategies. A prospector strategy

is generally adopted by innovative firms that are the first to market and achieve success and may be accompanied by risks and uncertainties since it involves new product and market development. On the other hand, a defender strategy is stable and allows for cost saving through economies of scale and thus, tends to have low risk and uncertainty. SMEs must consider a prospector strategy given their small size and short survival duration.

In this respect, some of the previous research works approach the leading strategy from the negative viewpoint. For example, in Choi et al. [7] study, leading strategies are likely to have tax avoidance opportunities due to new product and new market exploration. Also, a leading strategy is more likely to have higher supervisory intelligence, financial risk, and business risk than a defensive strategy. In addition, the leading strategy has considered the acceptability of uncertainty, the complexity and instability of organizational structure. Also, the leading strategy has lower asset security, faster growth pattern, unstable governance, and information asymmetry than defensive strategies. Leading strategies seem to be costly than defensive strategies due to innovation and risk pursuits from a long-term perspective [8].

Choi et al. [7] suggested that there is no substitute, so it is more sensitive to reputation than a defensive strategy and has a high possibility of earnings management related to financing and low profitability. Hogan et al. [9] found that leading firms are more likely to violate financial reporting than in defensive strategies. This is because leading corporations have greater risk, uncertainty, decentralized and temporary organizational structure than defensive ones. In addition, these strategies were expected to be the first in the market and would be different from defensive strategies that showed stable, careful and incremental growth patterns due to rapid and sporadic patterns.

Bentley et al. [10] presented empirical evidence that operating cash flow and return on investment are relatively low because leading firms are more likely to be exposed to profitability or financial distress than defenders. Therefore, there is a study to explain the leading strategy with a negative view in accounting field research. If the characteristics of the firm are suitable for the leading strategy among the management strategies, then if the firm conducts the leading strategy, the future performance can be created, which may reduce profit quality and capital cost. In this study, we considered the aspect of the firm which was not considered in these management strategies. Among these management strategies, leading strategies are suited to the characteristics of SMEs that are rapidly adapting to changes and have high growth rates. SMEs have a high level of business risk and financial risks and characteristics differ from those of large corporations [11].

Unlisted SMEs are have a short maintenance period, making it difficult to expect cost advantage benefits from learning and scale, low profit and cash holdings, and high financial risk. It is important for a leading company to secure a new distribution network. Leading firms need to be able to acquire distribution networks to gain advantage in competition with large corporations that are already in the mature stage to secure distribution and generate cost-based performance. In particular, pursuing a leading-edge strategy in the industrial sector can result in low competition and good performance [12]. This study will be meaningful in suggesting a positive aspect of the company against the negative view on the leading strategy in the accounting field. In addition, it seems to have been meaningful that the unlisted SMEs proposed that a leading-edge strategy would be advantageous in terms of a strategy that could compete with various growth and future performance comparisons compared with listed companies.

This study draws on Miles and Snow's [5,6] classification of business strategies into prospector or defender strategies to examine the influence of unlisted SMEs' business strategies on future performance and growth sustainability. The strategies are further examined on the basis of new product development, marketing efforts, growth patterns, production efficiency, capital structure, and organizational stability [10]. In doing so, this study aims to determine the potentially differing influences of new product development and marketing efforts on future performance.

A key contribution of this study is its expansion of accounting research exploring the relationship between business strategies and earnings quality, internal disclosure systems, and cost of capital and

its impact on future performance and growth sustainability. In particular, accounting studies have been described in terms of high capital costs, tax avoidance opportunities, profit adjustment, and low performance aspects of the leading strategy from a negative point of view. However, in this study, it is meaningful to verify the effect of a leading strategy as a strategy for creating future performance in the case of unlisted SMEs in accordance with the characteristics of the company. This implies that it is meaningful to suggest a positive aspect of the leadership style strategy. In addition, these studies were mainly limited to small and medium-sized enterprises, unlike the case of large corporations or publicly traded companies. Furthermore, it verifies the influence of each business strategy factor on future performance and growth sustainability and accordingly, highlights strategic factors that are key to unlisted SMEs' future performance and growth sustainability.

This study is different from prior research related to management strategy as an important factor influencing the future performance and growth potential. In addition, this analysis can take in pros and cons of the management strategy that were not considered in the prior study. In particular, the purpose of this study is to suggest that management strategies are related to innovation, new market development, marketing, and investment activities, and proper management strategies play an important role in corporate performance and future growth have. In addition, this study is meaningful in that the unlisted SMEs are more risky and financially difficult than listed companies, but present evidence that they can improve future competitiveness and grow through leading-edge strategies.

## 2. Literature Review and Hypothesis Development

Strategy was considered a military term before Chandler [13] first applied it in the context of business studies, defining it as the deployment of resources to achieve firm goals. Organizational theory classifies strategy into firm, business, and functional aspects and studies have explored business strategy from a business aspect. According to Porter's [4] well-known business strategy theories, a firm's business strategies can be largely divided into product differentiation and cost leadership. Product differentiation indicates that the strategy involves the creation of products and services through design, trademark images, and innovative technologies [14]. Cost leadership, on the other hand, entails cost reduction through economies of scale, learning through experience, and achieving low costs for goods and services by minimizing expenses related to research and development, services, and marketing.

Miles and Snow [5,6] classify firms into prospector and defender firms within a continuum with the analyzer firm existing somewhere in between these firm. A prospector strategy encourages research and development to seek new markets, products, and technologies so that firms can spearhead success as the first to market [15]. This strategy does not account for alternative goods or competitors. Furthermore, investors' expectations are low in terms of immediate financial gain but high regarding prominent changes and returns in the long term investments. The prospector strategy is characterized by quick growth, large variations in profits [2], distributed control systems, and nominal and weak control to encourage individual creativity and organizations' revolutionary members. However, this strategy requires long-term views on non-financial reliability and criticisms and involves widely varying performance indicators and a large degree of risks [1,10,16]. Moreover, it requires large cash outflows for research and development, human resources, and marketing and is dependent on external capital sources [10]. The defender strategy, on the other hand, aims at economy of capital and consistent performance when there are many alternatives and stability. This strategy minimizes marketing activities and seeks to achieve production efficiency. It also aims to achieve economies of scale and reduce production costs while maintaining stability and soundness within the range of expectations. The strategy is bureaucratic, has a centralized structure, and is characterized by strong control. Its key focus areas are slow change, stability, and objective measurement of production costs, productivity and efficiency, and present performance [10].

This study adopts Miles and Snow's [5,6] classification of prospector and defender strategies to determine their influence on unlisted SMEs' future. There have been various studies on

business strategies from various perspectives including earnings quality, executive compensation, financial reporting irregularities, tax aggressiveness, cost of capital, and internal control financial reporting [1,8,10,17–22].

Bentley et al. [23] found that management strategy is related to design and internal control systems. The internal control system is an important factor influencing the quality of financial reporting and therefore the management strategy will affect the quality of financial reporting. Leading strategies in management strategies are pioneering new markets, have fast growth rates, have complex internal control systems, and are highly volatile [2]. In contrast, defensive strategies pursue efficiency, are stable in production, and have stable growth patterns. Therefore, the defensive strategy is lower than the leading strategy and the growth or complexity level is lower than the leading strategy. The defensive strategy is dependent on short—term performance, but the leading strategy is not. In this study, we show that the leading strategy is high level of vulnerability of ICFR, and it is expected that the re—writing of financial information and litigation are likely to be high.

Chen and Jermias [24] find that firm's competitive strategy affects manager compensation. In this study, performance related compensation, manager's stock ownership, long-term compensation plan are explained as manager compensation, and long-term compensation plan has negative relation with performance. This study considered this relationship to be dependent on business strategy.

Houqe et al. [19] defined the purpose of corporate strategy as a leading, defensive, analytic reactor proposed by Snow and Hambrick [25]. The reason for distinguishing these corporate strategies is that it is a way to identify the implementation and realization of the strategy. In this study, we see that leading companies have a strong commitment to R&D and innovation. They frequently change products and markets and flourish in a changing business environment. Leading strategies are non-predictive and provide new market insights by constantly making new investments. In contrast, defensive strategies are stressed on the efficiency of sales, production is low and defensive. Therefore, companies pursuing defensive strategies view price, delivery, and quality as important competitors. Defensive corporations may be a difficult task for SMEs because they seek enormous efficiency rather than innovation, and they invest heavily in the establishment process and require huge funds.

The prospector strategy is defined by considerable growth options, rapid growth, brand creation, new market development, and active research and development, which indicates a low tendency toward present earnings [10,26,27] and the ability to achieve future profits. The defender strategy, by contrast, involves slow and low growth and emphasizes maintaining and safeguarding a firm and avoiding risk. This strategy tends to be dependent on present performance [10] and thus is unlikely to generate excess earnings in the future. Moreover, the defender strategy could hamper sustainable growth firms adopting this strategy will find it difficult to adapt to change given economies of scale. From the discussion thus far, it appears that the prospector strategy could be more successful in increasing future performance and growth potential, whereas the defender strategy is likely to reduce or maintain future performance or growth potential through cost cutting and lowered production costs.

The two business strategies are also influenced by the risks a firm faces. The prospector strategy has a high degree of future uncertainty and possibly higher operational risks owing to failure; moreover, present-day strategies may not be linked with future performance. On the other hand, the defender strategy allows for stable profit creation because it sustainably maintains its businesses approaches. Therefore, for firms relying on risks, implementing the defender strategy may be a better option than the prospector strategy to achieve future performance.

In general, expanding facility capacity and achieving cost reductions through economies of scale is more challenging for unlisted SMEs because as they are smaller than listed or larger firms. Moreover, unlisted SMEs tend to be associated with a shorter survival duration compared to larger firms, making it difficult to achieve cost reductions through learning. However, their smaller size and lower number of employees are an advantage because they allow unlisted SMEs to quickly adapt to change. In addition, they are able to increase performance through superior technology and core competencies. Therefore, the prospector strategy is more advantageous for unlisted SMEs to achieve future performance through

cost reductions. Although the prospector strategy's focus on present performance is a disadvantage, its active approach to investment activities such as the development of new products and markets will allow firms to generate excess profits in the future. In contrast, for firms chasing present performance through cost reductions and lower expenses a defender strategy is more beneficial, although associating this strategy with profits can be difficult. Moreover, the strategy involves reducing investments in the future and expenses tend to be postponed, hindering firms' ability to increase future profits.

Leading strategies in unlisted SMEs are already saturating industries or are not competitive and can increase future performance [12]. In addition, the greater the power of the chief executive officer (CEO), the more negative the impact on future performance [27]. The power of major creditors such as banks and financial institutions is strong for unlisted SMEs. If a leading strategy is an unethical or opportunistic act of a manager, a financial institution may put pressure on the firm. If the monitoring function is strengthened, it is possible that indiscriminate investments that deteriorate future performance may be reduced [28,29]. Therefore, the monitoring function can be strengthened in order to reduce the financial risk of the unlisted SMEs whose banks are pursuing a leading strategy with a relatively high level of business risk, which may result in a higher future performance than the defensive strategy.

**Hypothesis 1a.** *The prospector strategy more positively influences unlisted SMEs' future performance than the defender strategy.*

A firm's growth potential is closely related to its strategy. The literature on a firm's lifecycle classifies a firm's characteristics on the basis of its growth levels [27,30–35]. Accordingly, firms at a high-growth stage engage a diverse range of strategies including the development of new markets and technologies, marketing advertisement, and research and development. This approach can be seen as more similar to the prospector strategy than the defender strategy. Firms in their maturity and decline exhibit slow growth and focus on profit creation through cost-cutting, features which are highly related to the defender strategy. Smaller and newer firms, on the other hand, are considered to be in their growth stage and, thus, the prospector strategy is important for such firms to obtain ongoing growth potential.

**Hypothesis 1b.** *The prospector strategy has a more positive influence on unlisted SMEs' future growth potential than the defender strategy.*

This study also aims to determine business strategy indicators that influence unlisted SMEs' future performance and growth potential. The following six indicators are studied: research activities for new product development, production and distribution capabilities for product and service efficiency, investment opportunities, marketing for new product and service development, promotions, firm stability and decentralization, and production efficiency for capital investments. Among these measured variables, marketing and research and development activities are directly related to the development of new markets, technologies, and products and are likely to influence future performance and growth potential. Thus, the following hypothesis is proposed:

**Hypothesis 2.** *The influence of marketing efforts (or new product development) on unlisted SMEs' future performance and growth differs from that of other business strategy indicators.*

## 3. Methodology

This study uses the following models (1) and (2) to verify the influence of business strategies on unlisted SMEs' future performance and growth potential.

$$ROA_{i,t+k} = a_0 + a_1 SD_{i,t} + a_2 SIZE_{i,t} + a_3 LEV_{i,t} + a_4 Growth_{i,t} + a_5 ROA_{i,t} + a_6 Loss_{i,t} + a_7 ICF_{i,t} + a_8 Big4_{i,t} + YD_t + ID_{i,t} + e_{i,t} \tag{1}$$

where

$ROA_{i,t+k}$ is firm future performance in year $t + k$ determined by total returns in $t + k$ divided by total assets at the beginning of period (where $k = 1, 2, 3$);

$SD_{i,t}$ is an indicator that equals 1 for prospector firms and 0 for defender firms;

$SIZE_{i,t}$ is a natural log value for firm total assets in year $t$ and denotes firm size in year $t$;

$LEV_{i,t}$ is the ratio of total debt in year $t$ divided by total assets in period $t$ and indicates financial risk faced by firm $i$ in year $t$;

$Growth_{i,t}$ is firm sustainable growth in year $t$ measured using sales growth (i.e., sales in year $t - 1$ is subtracted from sales in year $t$ and then divided by sales in year $t - 1$);

$ROA_{i,t}$ is the net earnings on firm total assets in year $t$ and denotes the ratio obtained from dividing net earnings in year $t$ by total assets at the beginning of the period;

$Loss_{i,t}$ is an indicator that equals 1 if the firm reports losses in period $t$ and 0 if not negative;

$ICF_{i,t}$ is firm cash holdings in year $t$ and is calculated by dividing cash and cash equivalents in year $t$ by total assets in period $t$; $Big4_{i,t}$ is an indicator that equals 1 if firm auditor in year $t$ is a Big 4 accounting firm and 0 otherwise; $YD_t$ is a control variable for the year of analysis that takes the value of 1 if the present year is the year of analysis and 0 otherwise; and

$ID_{i,t}$ is an industry control variable for firm $i$ in year $t$ that takes the value of 1 if the firm belongs to the industry and 0 otherwise.

In Equation (1), future performance is measured using return on assets (ROA) from year $t + 1$ to year $t + 3$. If the business strategy is a prospector and positively influences the firm's future performance, the regression coefficient will be $a_1 > 0$. This study uses size (*SIZE*) as a control variable because a greater size allows for easier access to external capital. However, larger firms find it more difficult to implement changes and may incur high fixed expenses, thus negatively influencing future performance. A higher leverage ratio (*LEV*) renders funding access challenging and results in higher proportions of interest expenses, thus negatively influencing future performance. Earnings and growth tend to be continuous. To control for these factors, this study measures present performance and growth potential as present-period ROA and accounts for present sales growth (*Growth*) in the model. In the case of losses, firms may struggle to access funding or experience reduced earnings owing to the increased costs of third-party capital. Thus, this study includes a loss dummy variable (*Loss*) in the model. Greater internal cash holdings allow a firm to maintain business continuity and higher liquidity. Such firms can link investment opportunities with performance. Therefore, the level of cash and cash equivalents are used as a control variable. Audit firms (*Big4*) determine audit quality—the higher the quality, the lower the present business earnings—and, thus, are included in the model because they are expected to reduce cases of increasing present-period performance by reducing future performance. To control for the influence of industry and time, an industry (ID) and year (YD) dummy are included as control variables. Finally, future growth potential is measured using sales growth (*Growth*) from $t + 1$ to $t + 3$ and is presented in the following equation:

$$Growth_{i,t+j} = a_0 + a_1 SD_{i,t} + a_2 SIZE_{i,t} + a_3 LEV_{i,t} + a_4 Growth_{i,t} + a_5 ROA_{i,t} + a_6 Loss_{i,t} + a_7 ICF_{i,t} + a_8 Big4_{i,t} + YD_t + ID_{i,t} + e_{i,t} \tag{2}$$

where

$Growth_{i,t+j}$ is sustainable growth of in year $t + j$ measured using sales growth. For example, if $j = 1$, sales in year $t$ is subtracted from sales in year $t + 1$ and divided by sales in year $t$.

In Equation (2), if the business strategy positively influences future growth potential, regression coefficient $a_1$ will have a significant positive value. This study formulates Equations (3) and (4) to verify the influence of new product development and marketing efforts on future performance and growth potential.

$$ROA_{i,t+k} = a_0 + a_1 Str_{p,i,t} + a_2 SIZE_{i,t} + a_3 LEV_{i,t} + a_4 Growth_{i,t} + a_5 ROA_{i,t} + a_6 Loss_{i,t} + $$
$$a_7 ICF_{i,t} + a_8 Big4_{i,t} + YD_t + ID_{i,t} + e_{i,t} \tag{3}$$

$$Growth_{i,t+j} = {}^a0 + a_1 Str_{p,i,t} + a_2 SIZE_{i,t} + a_3 LEV_{i,t} + a_4 Growth_{i,t} + a_5 ROA_{i,t} + a_6 Loss_{i,t} + $$
$$a_7 ICF_{i,t} + a_8 Big4_{i,t} + YD_t + ID_{i,t} + e_{i,t} \tag{4}$$

where

$Str_{p,i,t}$ is the measurement variable for firm business strategy in year $t$. $p = 1$ indicates the level of marketing efforts and is measured by dividing selling, general, and administrative expenses by sales. $p = 2$ is the indicator for new product development and is measured by dividing research and development expenses by sales.

If new product development and marketing efforts influence future performance and growth potential, the regression coefficient $a_1$ will be significant.

Table 1 presents the approach to measuring a business strategy, a variable of interest in this study. The following business strategy is classified into defender and prospector strategies and measured using their method. In addition, this analysis adopts the methodology of Ryu et al. [8,18] to study Korean firms.

**Table 1.** Approach to measuring business strategies [15].

| Business Strategy Variable | Measuring Approach |
| --- | --- |
| $Str_1$ (marketing efforts) | selling, general, and administrative expenses/sales |
| $Str_2$ (new product development) | research and development expense/sales |
| $Str_3$ (production efficiency) | employees/sales |
| $Str_4$ (capital structure) | net property, plant, and equipment/total assets |
| $Str_5$ (organizational stability) | standard deviation of total number of firm employees |
| $Str_6$ (growth pattern) | percentage change in sales |

Using the above equation, each item is measured over a five-year period and uses a five-year average ratio. The values are then classified into years and industries and form five groups, where the lowest value is 1 and highest is 5. Since capital intensity is likely to be high for a defender strategy, which seeks cost reductions through economies of scale, the values are reversed. Adding these values produces a score between 6 and 30. Accordingly, firms with a score between 6 and 13 denote a defender and those with a score from 24 to 30 are considered a prospector. Firms that score between the ranges, that is, between 14 and 23, are viewed as analyzers. This study analyzes for prospectors and defenders on the basis of total scores and conducts an additional analysis to determine if differences in business strategies influence the organizational lifecycle.

## 4. Data Analysis and Results

### 4.1. Sample Selection

This study uses data on unlisted SMEs in Korea from the KIS (Korea Information Service)-Value Database. Table 2 discusses the sample selection. Unlisted SMEs excluded 105,658 actual samples because it is difficult to obtain more data than listed companies or large corporations. Also, 75,751 samples corresponding to the analytical strategy, which is an intermediate process, were excluded from the analysis to verify the difference between the leading and defensive strategies. So, the total number of samples we used in our analysis is 20,216 (firms-years).

**Table 2.** Sample selection.

| Description | Firm-Years |
|---|---|
| Firm years for unlisted small and medium-sized enterprises (SMEs) in KIS (Korea Information Service)-Value Database | 212,625 |
| Firm years with incomplete data are excluded | (105,658) |
| Firms with analytical strategies are excluded | (75,751) |
| Final sample | 20,216 |

As shown in Table 2, the analysis includes 20,216 firms (firm years) with available data on the KIS (Korea Information Service)-Value database to identify prospectors and defenders.

*4.2. Descriptive Statistics and Correlations*

Table 3 indicates that the average score for business strategy (*Str*) is 15.420 and the average marketing efforts (*Str*$_1$) is 8% of total sales. Research and development expenses (*Str*$_2$) are approximately 0.8% of total sales, which is about one-tenth of selling, general, and administrative expenses. This value is lower than that for listed companies, indicating that financial difficulties could lead to shortfalls in resource allocation for new product development. ROA, an alternative for future performance, ranges between 0.04% and 0.05%. The average future growth potential (*Growth*) is near 10%, which is higher than that for listed firms in Korea. Average firm size (*SIZE*) and leverage (*LEV*) are 23.541 and 0.429. Internal cash holdings (*ICF*) are 0.067, which is 6.7% of total assets. The average loss dummy (*Loss*) is 0.136, suggesting that 13.6% of all analyzed samples report losses. Finally, only 11% firms have been audited by a Big 4 accounting firm. In other words, audit quality is considerably lower than that of listed firms, which is 70%–80%.

**Table 3.** Descriptive statistics [a].

| Variable | Mean | Medium | Q1 | Q3 | Std. Dev. |
|---|---|---|---|---|---|
| $Str_1$ | 0.082 | 0.037 | 0.021 | 0.037 | 0.121 |
| $Str_2$ | 0.008 | 0.000 | 0.000 | 0.000 | 0.027 |
| $ROA_{i,t+1}$ | 0.051 | 0.036 | 0.009 | 0.085 | 0.122 |
| $ROA_{i,t+2}$ | 0.047 | 0.034 | 0.008 | 0.080 | 0.114 |
| $ROA_{i,t+3}$ | 0.043 | 0.032 | 0.007 | 0.076 | 0.111 |
| $Growth_{i,t+1}$ | 0.122 | 0.045 | −0.089 | 0.220 | 0.561 |
| $Growth_{i,t+2}$ | 0.125 | 0.044 | −0.086 | 0.215 | 0.589 |
| $Growth_{i,t+3}$ | 0.099 | 0.033 | −0.096 | 0.189 | 0.564 |
| $SIZE_{i,t}$ | 23.541 | 23.552 | 23.023 | 24.061 | 0.904 |
| $LEV_{i,t}$ | 0.429 | 0.430 | 0.194 | 0.077 | 0.850 |
| $ROA_{i,t}$ | 0.064 | 0.040 | 0.011 | 0.091 | 0.134 |
| $Growth_{i,t}$ | 0.157 | 0.019 | −0.121 | 0.201 | 1.929 |
| $ICF_{i,t}$ | 0.067 | 0.030 | 0.008 | 0.086 | 0.096 |
| $Loss_{i,t}$ | 0.136 | 0.000 | 0.000 | 0.000 | 0.343 |
| $Big4_{i,t}$ | 0.110 | 0.000 | 0.000 | 0.000 | 0.313 |

*Str* $_{p,i,t}$ measurement variable for firm business strategy in year t (*p* = 1–2); *ROA*$_{i,t+k}$ future performance in year *t* + *k* (where *k* = 1, 2, 3); *Growth*$_{i,t+j}$ sustainable growth in *t* + *j* (where *j* = 1, 2, 3); *SD*$_{i,t}$ an indicator that equals 1 for prospector firms and 0 for defender firms; *SIZE*$_{i,t}$ firm size of firm *i* in year *t*; *LEV*$_{i,t}$ the financial risk that firm faces in year *t*; *Growth*$_{i,t}$ sustainable growth in year *t* and is measured using sales growth; *ROA*$_{i,t}$ the net income on firm total assets in year *t* and denotes the ratio obtained from dividing net earnings in year *t* in total assets at the beginning of the period; *Loss*$_{i,t}$ an indicator that equals 1 if the firm reports loss in period *t* and 0 if not negative; *ICF*$_{i,t}$ cash holdings in year *t*; *Big4*$_{i,t}$ an indicator that equals 1 if the auditor of firm *i* in year *t* is a Big 4 accounting firm and 0 otherwise.

Table 4 reports the results of Pearson's correlation analysis for the major variables. The results indicate a significant positive relationship among the dependent variable's future performance, total ROA, and sales growth (r = 0.251, $p < 0.001$).

**Table 4.** Pearson's correlation among variables (n = 20,216) [a].

| | $ROA_{i,t+1}$ | $Growth_{i,t+1}$ | $SD_{i,t}$ | $Str_{1,i,t}$ | $Str_{2,i,t}$ | $SIZE_{i,t}$ | $LEV_{i,t}$ | $ROA_{i,t}$ | $Growth_{i,t}$ | $ICF_{i,t}$ | $Loss_{i,t}$ |
|---|---|---|---|---|---|---|---|---|---|---|---|
| $Growth_{i,t+1}$ | 0.251 (0.000) | | | | | | | | | | |
| $SD_{i,t}$ | 0.173 (0.000) | 0.162 (0.000) | | | | | | | | | |
| $Str_{1,i,t}$ | −0.002 (0.731) | 0.094 (0.000) | 0.369 (0.000) | | | | | | | | |
| $Str_{2,i,t}$ | 0.088 (0.000) | 0.087 (0.000) | 0.438 (0.000) | 0.141 (0.000) | | | | | | | |
| $SIZE_{i,t}$ | −0.238 (0.000) | −0.190 (0.000) | −0.259 (0.000) | 0.034 (0.000) | −0.181 (0.000) | | | | | | |
| $LEV_{i,t}$ | −0.205 (0.000) | 0.053 (0.000) | −0.110 (0.000) | 0.007 (0.333) | −0.116 (0.000) | −0.032 (0.000) | | | | | |
| $ROA_{i,t}$ | 0.426 (0.000) | 0.050 (0.000) | 0.226 (0.000) | 0.024 (0.001) | 0.092 (0.000) | −0.207 (0.000) | −0.268 (0.000) | | | | |
| $Growth_{i,t}$ | 0.124 (0.000) | 0.160 (0.000) | 0.292 (0.000) | 0.101 (0.000) | 0.103 (0.000) | −0.168 (0.000) | 0.043 (0.000) | 0.234 (0.000) | | | |
| $ICF_{i,t}$ | 0.224 (0.000) | 0.041 (0.000) | 0.228 (0.000) | 0.058 (0.000) | 0.168 (0.000) | −0.167 (0.000) | 0.332 (0.000) | 0.239 (0.000) | 0.087 (0.000) | | |
| $Loss_{i,t}$ | −0.215 (0.000) | 0.010 (0.000) | −0.078 (0.000) | 0.083 (0.000) | 0.001 (0.833) | 0.118 (0.000) | 0.175 (0.000) | 0.409 (0.000) | −0.040 (0.000) | −0.117 (0.000) | |
| $Big4_{i,t}$ | 0.018 (0.012) | −0.012 (0.083) | 0.070 (0.000) | 0.057 (0.000) | 0.031 (0.000) | 0.239 (0.000) | −0.089 (0.000) | −0006 (0.376) | −0.007 (0.306) | 0.085 (0.000) | 0.050 (0.000) |

[a] Refer to the comments in Table 3.

Business strategy (*SD*) has a significant positive influence on future performance (r = 0.173, *p* < 0.001) and future growth potential (r = 0.162, *p* < 0.001). In addition, marketing efforts (*Str*₁) has a significant positive influence on future growth sustainability, although this is not the case for future performance.

New product development (*Str*₂) reports a significant positive relationship with future performance and future growth potential (r = 0.088, *p* < 0.001; r = 0.087, *p* < 0.001).

Firm size (*SIZE*) and leverage (*LEV*), on the other hand, are negatively related with future performance. Present-period performance (*ROA*), growth (*Growth*), and internal cash holdings (*ICF*) have significant positive relationships with future performance. Finally, future performance and loss (*Loss*) have a negative relationship, whereas auditor (*Big4*) is positively associated with future performance.

### 4.3. Hypothesis Testing

Table 5 presents the results for the influence of a firm's business strategy on future performance and growth potential. First, business strategy (*SD*) positively influences future performance; in other words, the prospector strategy employed by unlisted SMEs contributes more positively to their future performance than the defender strategy. In addition, business strategy (*SD*) influences a firm's future growth potential. Thus, the prospector strategy has a significant positive influence on future growth potential. It can be concluded that business strategy plays an important role in future performance and growth potential.

Firm size (*SIZE*), on the other hand, negatively influences future performance or sustainable growth potential. This result indicates that larger sizes are more open to change, new challenges, and innovations difficulties and thus, firm size may have weak relationships with future performance and growth potential. Higher leverage ratio (*LEV*) hinders funding access and this affects day-to-day business activities. Cost of third-party capital such as interest expenses may decrease future performance. Thus, leverage ratio has a significant negative impact on future performance. However, these results are not consistent for sustainable growth potential. While leverage ratio increases next-period sales, it is insignificant thereafter, and thus, it is difficult to confirm the influence of leverage ratio on sustainable growth potential. Present-period performance has a significant positive relationship with future performance, but this relationship is not consistent for growth potential. Present-period sales growth (*Growth*) has an insignificant relationship with future performance; however, it has a significantly

positive relationship with growth potential. Internal cash holdings (*ICF*) have a significant positive relationship with future performance, although its influence on future growth could not be confirmed. The loss dummy reports a significant negative influence on future performance but a significant positive influence on growth potential. These results indicate that firms in their growth stage and with high growth potential may report negative earnings rather than positive profits given their proactive investment activities. Moreover, the audit dummy (*Big4*) has a significant positive influence on future performance and growth potential, indicating an increase in earnings quality and decrease in over-estimated present-period profits and under-estimated next-period profits.

**Table 5.** Regression of future performance ($ROA_{i,t+k}$) or growth sustainability ($Growth_{i,t+j}$) on business strategy ($SD_{i,t}$) [a].

| Variables | Prediction | Results | | | | | |
|---|---|---|---|---|---|---|---|
| | | Regression 1 (with $ROA_{i,t+k}$) | | | Regression 2 (with $Growth_{i,t+j}$) | | |
| | | $ROA_{i,t+1}$ | $ROA_{i,t+2}$ | $ROA_{i,t+3}$ | $Growth_{i,t+1}$ | $Growth_{i,t+2}$ | $Growth_{i,t+3}$ |
| | | Coefficients | Coefficients | Coefficients | Coefficients | Coefficients | Coefficients |
| *Intercept* | ? | 0.568 *** | 0.630 *** | 0.549 *** | 2.216 *** | 2.806 *** | 1.952 *** |
| $SD_{i,t}$ | + | 0.010 *** | 0.007 *** | 0.005 ** | 0.152 *** | 0.041 *** | 0.022 ** |
| $SIZE_{i,t}$ | +/− | −0.021 *** | −0.024 *** | −0.020 *** | −0.092 *** | −0.113 *** | −0.080 *** |
| $LEV_{i,t}$ | − | −0.050 *** | −0.049 *** | −0.050 *** | 0.159 *** | −0.021 | 0.004 |
| $ROA_{i,t}$ | + | 0.297 *** | 0.171 *** | 0.112 *** | −0.014 | −0.093 ** | 0.055 |
| $Growth_{i,t}$ | + | 0.001 | −0.001 | −0.003 *** | 0.062 *** | 0.019 *** | 0.025 *** |
| $ICF_{i,t}$ | + | 0.076 *** | 0.061 *** | 0.041 *** | 0.007 | −0.086 * | −0.018 |
| $Loss_{i,t}$ | − | −0.014 *** | −0.014 *** | −0.012 *** | 0.053 *** | 0.058 *** | 0.066 *** |
| $Big4_{i,t}$ | + | 0.003 | 0.009 *** | 0.008 *** | 0.045 *** | 0.047 *** | 0.014 |
| *Adj. $R^2$* | | 0.226 | 0.145 | 0.102 | 0.080 | 0.045 | 0.034 |
| *Max. VIF* | | 1.520 | 1.520 | 1.520 | 1.520 | 1.520 | 1.520 |
| *Sample size* | | 20,216 | 20,216 | 20,216 | 20,216 | 20,216 | 20,216 |

\* $p < 0.05$, \*\* $p < 0.01$, \*\*\* $p < 0.001$, [a] Refer to the comments in Table 3.

Table 6 presents the results for the influence of marketing efforts as a business strategy segment, denoted by selling, general, and administrative expenses divided by sales in terms of future performance and sustainability growth.

The results show that marketing efforts ($Str_1$) negatively influences next-period performance but does not influence performance thereafter; however, marketing efforts have a significant positive influence on sustainable growth potential. A firm's marketing expenses increase future sales volume, thus indicating an increase in sales. Nevertheless, the influence on future performance is unclear given the increased expenses.

Table 7 reports the influence of new product development, as denoted by research and development, on future performance and growth potential. Research and development activities have a significant positive influence on future performance and growth potential. Therefore, this study can confirm that SMEs' innovative activities, and particularly the development of new products and technologies, strengthen a firm's core competencies and positively contribute to future performance or growth potential.

**Table 6.** Regression of future performance ($ROA_{i,t+k}$) and growth sustainability ($Growth_{i,t+j}$) on marketing efforts ($Str_{1,i,t}$) [a].

| Variables | Prediction | Results | | | | | |
|---|---|---|---|---|---|---|---|
| | | Regression 1 (with $ROA_{i,t+k}$) | | | Regression 2 (with $Growth_{i,t+j}$) | | |
| | | $ROA_{i,t+1}$ | $ROA_{i,t+2}$ | $ROA_{i,t+3}$ | $Growth_{i,t+1}$ | $Growth_{i,t+2}$ | $Growth_{i,t+3}$ |
| | | Coefficients | Coefficients | Coefficients | Coefficients | Coefficients | Coefficients |
| *Intercept* | ? | 0.593 *** | 0.644 *** | 0.559 *** | 2.431 *** | 2.832 *** | 1.969 *** |
| $Str_{1,i,t}$ | + | −0.029 *** | −0.001 | 0.001 | 0.456 *** | 0.219 *** | 0.144 *** |
| $SIZE_{i,t}$ | +/− | −0.021 *** | −0.024 *** | −0.021 *** | −0.101 *** | −0.115 *** | −0.081 *** |
| $LEV_{i,t}$ | − | −0.051 *** | −0.049 *** | −0.050 *** | 0.073 *** | -0.026 | 0.001 |
| $ROA_{i,t}$ | + | 0.299 *** | 0.172 *** | 0.113 *** | 0.015 | −0.086 ** | 0.059 * |
| $Growth_{i,t}$ | + | 0.003 *** | −0.000 | −0.003 *** | 0.073 *** | 0.020 *** | 0.026 *** |
| $ICF_{i,t}$ | + | 0.081 *** | 0.064 *** | 0.043 *** | 0.058 | −0.075 | −0.011 |
| $Loss_{i,t}$ | − | −0.013 *** | −0.014 *** | −0.012 *** | 0.041 *** | 0.051 *** | 0.062*** |
| $Big4_{i,t}$ | + | 0.005 * | 0.009 *** | 0.008 *** | 0.054 *** | 0.047 *** | 0.014 |
| *Adj. $R^2$* | | 0.226 | 0.144 | 0.101 | 0.077 | 0.047 | 0.034 |
| *Max. VIF* | | 1.447 | 1.447 | 1.447 | 1.447 | 1.447 | 1.447 |
| *Sample size* | | 20,216 | 20,216 | 20,216 | 20,216 | 20,216 | 20,216 |

\* $p < 0.05$, \*\* $p < 0.01$, \*\*\* $p < 0.001$, [a] Refer to the comments in Table 3.

**Table 7.** Regression of future performance ($ROA_{i,t+k}$) and growth sustainability ($Growth_{i,t+j}$) on new product development ($Str_{2,i,t}$) [a].

| Variables | Prediction | Results | | | | | |
|---|---|---|---|---|---|---|---|
| | | Regression 1 (with $ROA_{i,t+k}$) | | | Regression 2 (with $Growth_{i,t+j}$) | | |
| | | $ROA_{i,t+1}$ | $ROA_{i,t+2}$ | $ROA_{i,t+3}$ | $ROA_{i,t+1}$ | $Growth_{i,t+2}$ | $Growth_{i,t+3}$ |
| | | Coefficients | Coefficients | Coefficients | Coefficients | Coefficients | Coefficients |
| *Intercept* | ? | 0.582 *** | 0.631 *** | 0.550 *** | 2.412 *** | 2.846 *** | 1.967 *** |
| $Str_{2,i,t}$ | + | 0.056 * | 0.133 *** | 0.099 *** | 1.145 *** | 0.412 ** | 0.315 ** |
| $SIZE_{i,t}$ | +/− | −0.021 *** | −0.024 *** | −0.020 *** | −0.100 *** | −0.115 *** | −0.081 *** |
| $LEV_{i,t}$ | − | −0.050 *** | −0.048 *** | −0.050 *** | 0.160 *** | −0.020 | 0.005 |
| $ROA_{i,t}$ | + | 0.299 *** | 0.172 *** | 0.113 *** | 0.017 | −0.085 ** | 0.060 * |
| $Growth_{i,t}$ | + | 0.002 | −0.001 | −0.003 *** | 0.076 *** | 0.022 *** | 0.027 *** |
| $ICF_{i,t}$ | + | 0.079 *** | 0.061 *** | 0.041 *** | 0.043 | −0.078 | −0.015 |
| $Loss_{i,t}$ | − | −0.014 * | −0.01 *** | −0.012 *** | 0.048 *** | 0.057 *** | 0.065 *** |
| $Big4_{i,t}$ | + | 0.004 | 0.009 *** | 0.008 *** | 0.058 *** | 0.050 *** | 0.015 |
| *Adj. $R^2$* | | 0.225 | 0.145 | 0.102 | 0.073 | 0.045 | 0.034 |
| *Max. VIF* | | 1.967 | 1.520 | 1.520 | 1.520 | 1.520 | 1.520 |
| *Sample size* | | 20,216 | 20,216 | 20,216 | 20,216 | 20,216 | 20,216 |

\* $p < 0.05$, \*\* $p < 0.01$, \*\*\* $p < 0.001$, [a] Refer to the comments in Table 3.

Although, not covered in the table, capital structure ($Str_4$) as a business strategy aspect negatively influences future performance and growth sustainability. It was expected that larger capital expenditures, such as for production facilities, would increase performance given the economies of scale; however, these results remain unconfirmed for unlisted SMEs. Even if some unlisted SMEs have greater proportions of production equipment and larger capital expenditures, they remain at levels lower than those of large enterprises, thus limiting the possibility of decreasing production cost. This, in turn, could lead to higher proportions of fixed production costs, which will negatively influence future performance. Organizational stability ($Str_5$), measured using employee turnover, has a negative influence on future performance and higher proportions of labor costs could decrease performance in the near future. Growth pattern ($Str_6$) has a significant positive relationship with growth potential

rather than future performance. Finally, production efficiency ($Str_3$) does not have a consistent relationship with future performance or growth potential.

## 5. Discussion

This study measured the management strategy as a leading and defensive strategy of Miles and Snow [5,6] and verified the effect of the strategic difference on the future performance and growth potential of the unlisted SMEs. If the management strategy is measured and analyzed by using other methods, the research result may be different from this research. Therefore, there is a need to measure management strategies through more detailed and accurate methods in the future. The results of this study are difficult to generalize to the analysis of small and medium enterprises in Korea. Therefore, there is a need to analyze more countries in the future.

We prove that the sample can be a better model if we verify the difference of results when the same company conducts different strategies. In addition, if we measure sustainability through other measures besides future performance and profits and sales growth, we can secure the robustness of our research results. Despite these limitations, this study suggests a leading strategy as a more suitable management strategy for unlisted SMEs, and suggests that it can be applied to business strategy considering cost reduction or structural problems.

## 6. Conclusions

Drawing on Miles and Snow [5,6], this study classifies unlisted SMEs' business strategies into prospector and defender strategies and examines their influence on future performance and growth potential with a dataset derived from Korea. In addition, it analyzes the influence of marketing efforts and new product development on future performance and growth sustainability. To do so, it uses data for unlisted Korean SMEs between 2006 and 2014 and obtains the following findings.

First, compared to the defender strategy, the prospector strategy has a more significant positive relationship on unlisted SMEs' future performance and growth potential. In other words, being an unlisted SME is a favorable condition for innovation, new market development, and change and since the prospector strategy is closely associated with these characteristics, it positively contributes to future performance and growth potential.

Second, there is no direct relationship between marketing efforts, a measure of business strategy, and future performance in unlisted SMEs; nevertheless, marketing efforts play a key role in increasing growth potential and future sales.

Finally, investments in research and development have a significant positive influence on future performance and growth potential. Thus, research and development increase the unlisted SMEs' competencies to create change, innovation, and differentiated performance, thereby contributing to future performance and growth potential.

In the field of accounting related to management strategy, it is suggested that leading strategy leads to high capital cost and low profitability. In this study, however, it is meaningful to suggest that leading strategies of unlisted SMEs are more likely to help future performance or growth than defensive strategies. These results are likely to be meaningful for small firms, not for all companies. In addition, unlisted SMEs may be the result of enhanced sanctions or monitoring, screening, and oversight of financial institutions, while other shareholders are less capable of monitoring.

Malesios et al. [36] analyzed SMEs with an interest in social and environmental factors, business practice and financial performance, and found that the relationship between sustainability and financial performance was significantly positive. The results show that the social, environmental, and practical aspects are directly related to firm sustainability, and they have a positive effect on future performance as a direct influence on management strategy. This study is expected to be meaningful in extending these studies and directly verifying the effects of management strategies on financial performance with high relevance to sustainability.

The significance of this study lies in its review of the relationship among business strategy, future performance, and growth potential. Unlisted SMEs, which are smaller in size and have shorter survival periods, should adopt strategies that focus on increasing their competencies for change, innovation, and the development of new products and markets, which could critically increase future sustainability. On the other hand, a defender strategy is more appropriate for large-scale enterprises seeking economies of scale through production facilities and cost reductions.

In this study, the business strategy has been verified by distinguishing between the leading type and the defensive type, but it is necessary to analyze it using other strategies in the future. In addition to the size and unlisted characteristics of the company, it is expected that the research should be expanded in the future.

This study can be regarded as a management activity that can affect overall innovation, technology, and marketing throughout the enterprise. Therefore, it can be regarded as an important factor affecting future competence beyond innovation and technology. In addition, this topic suggests a practical and sustainable way for the company to grow in the future.

**Author Contributions:** All authors contributed equally to the paper, namely to review the literature, collect data, apply research methods and interpret the results. Conceptualization, W.P. and C.G.B.; Data curation, W.P. and C.G.B.; Formal analysis, W.P.; Methodology, W.P.; Supervision, C.S.S.; Validation, C.S.S.; Visualization W.P.; Writing-review and editing, W.P. and C.G.B.

**Funding:** This work was supported by Uiduk University Foundation Grant, 2018.

**Conflicts of Interest:** The authors declare no conflict of interest.

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
