# Peer review of "Impact of Unlisted Small and Medium-Sized Enterprises’ Business Strategies on Future Performance and Growth Sustainability"

_2199-8531, doi:10.3390/joitmc5030060_

Round 1

Reviewer 1 Report

My comments on the paper - Impact of Unlisted Small and Medium-Sized  Enterprises’ Business Strategies on Future  Performance and Growth Sustainability- are as follows.

- The goal of the paper and main results are included in the abstract

- Keywords are appropriate

- The structure of the paper is appropriate

- The introduction provides the necessary background information and indicate contribution of the paper to the academic literature.

- In the introduction section, the structure of the paper on sections is not provided.

- The research methodology used by the author is adequate for the approached subject

- The results of the research are clearly underlined.

- It is not pointed out if the results of the research are in accordance or not with other studies. We recommend that the results obtained from the study should be compared with the results obtained in the case of similar researches from the academic literature.

- In section 4, the author does not indicate the time period for which the sample of firms is analyzed.

- There are some typing errors  in the paper (Line 414, 415)

- References: follow the journal guidelines

-  We consider that the author can show the limitations of the analysis carried out in his paper.

Author Response

Thank you for your kind review of the paper. The modifications are as follows.

Contribution of current manuscript to the related literature

→ Correct or supplement(in Conclusion).

Real importance and potential application of the results.

→ Correct or supplement(in Conclusion).

The recommendation for future studies.

→ Correct or supplement(in Conclusion).

Reviewer 2 Report

1. The structure of the paper needs attention and the usual rule
(introduction-rationale-need for the work/research questions,
background-literature review, approach-methods-research performed,
results, discussion and then conclusions/concluding remarks) should be
followed more closely to facilitate the flow of the paper. Please
develop further / expand your discussion of findings perhaps by drawing
on relevant studies and in relation with prior MDPI's-JOItmC literature -
develop further and expand your final section of concluding remarks;
incorporate research and policy recommendations in the final conclusion
section. Cite (primarily) in these final-most critical sections of your
manuscript relevant papers published in the Journal you submitted your
work to (in order to provide some sort of continuity of the specific
research string).

2. More references to relevant literature/empirical studies on innovation and/or sustainability perspectives  relevant to SMEs could increase the quality of the research paper and provide a much clearer message to the reader - these may perhaps help you building both your 'mapping of the issue under investigation' in the opening paragraphs as well as your discussion/concluding remarks which need to be extended. In this respect, I suggest adding the following to your reference list:

Halkos, G., & Skouloudis, A. (2018). Corporate social responsibility and innovative capacity: Intersection in a macro-level perspective. Journal of cleaner production, 182, 291-300.

Halkos, G., ... (2018). Bouncing Back from Extreme Weather Events: Some Preliminary Findings on Resilience Barriers Facing Small and Medium‐Sized Enterprises. Business Strategy and the Environment, 27(4), 547-559.

Malesios, C., ... (2018). Impact of small‐and medium‐sized enterprises sustainability practices and performance on economic growth from a managerial perspective: Modeling considerations and empirical analysis results. Business Strategy and the Environment, 27(7), 960-972.

3. The introductory/opening section should communicate a little clearer
the literature gaps, as well as the study's aims & objectives in
order to facilitate the flow of the study.

4. Concluding remarks – authors must elaborate more on what is their
contribution to the literature as well as on opportunities for future
research. Questions that need to be answered: Why your study is
important? and how it extendso existing knowledge on the issue/topic?
Conclusions need to be written in a clear and coherent manner and draw
the main lessons from the paper. I suggest you to concentrate on the
description of the implications of the work, the main findings and its
potential replicability - empirical investigation - elsewhere. Furthermore,
limitations of the study need to be outlined to a greater extent, and so
are any potential connections between your study and specific aspects
of the Journal's scope (i.e. innovation vis-a-vis sustainability perhaps).

5. Carefully check the references, so as to make sure they are all complete and follow the Guidelines to Authors.

6. Finally, when you submit the corrected version, please do check
thoroughly, in order to avoid grammar, syntax or structure/presentation
flaws. Make sure you retain a formal/academic-specific style of
presenting your work throughout the text - (if necessary) please seek
for professional English proofreading services or ask a native
English-speaking colleague of yours in order to refine and improve the
English in your paper.

Reviewer 3 Report

The manuscript is interesting and the remarks insightful, but the narrative flow has been structured in a vague manner, while there is weak connectivity of the results to other developing economies, and abstracted manipulation of the main entities noted at it title. Other shortcomings of the analysis are its vagueness, the lack of diagrammatic/schematic/graphical connectivity and depiction among those key-contributions upon “growth sustainability”, “profitability”, “innovativeness”, “Research and Development”, and “Future Performance”, explored. Therefore, the manuscript can be published after the consideration of the review comments suggested.

 1) The problem setting by authors is vaguely denoted at the “Abstract” section, even though the correlation of SMEs to growth sustainability and innovativeness is an attractive topic of wider socio-economic impact. Therefore, it is recommended authors to specify which are their place, time, and business sector(s) of analysis? The country of analysis, Korea, cannot be firstly referred at the middle of the research, p. 6/14, but at the “Abstract” section. The same comment applies to the period of analysis: it cannot be firstly denoted at the “5. Conclusions” section (p.12/14), but at the “Abstract” section.

2) Besides, the following statement at the “Abstract” section needs a more precise and quantitative explanation:  “…….SMEs which are small (here there is repetition, since the “S” from “SMEs” means “small”), financially challenged, and have shorter lifespans and faster growth rates…..” “shorter and faster”…..than what? Here it has been given a clarification.

 3) Again the argument that “This study is meaningful because it identifies the role of research and development in increasing future growth sustainability in SMEs” is hard to follow, since “research” and “development” are abstracted terms which are not always linked or directly perceived to “growth” and “profitability”. Therefore, it is recommended authors:

 a) At the results and discussion structure, the specific notation and brief explanation of the contribution of: financial status/pricing, technological advancements’ spur, governmental policies, extroversion and internationalization perspectives to an international and liberalized marketplace.

 b) Authors are recommended to determine which quantitative and qualitative advantages, in numerical data, are anticipated from their analysis, as well as to specify feasible measures that could be undertaken in order to achieve them.

 4) The following text extract is abstracted, therefore it has to be revised in a more meaningful manner:

 “In this study, we considered the aspect of the firm which was not considered in the previous study. Among these management strategies, leading strategies are suited to the characteristics of SMEs that are rapidly adapting to changes and have high growth rates.”

 Which is the “previous study”? Which are “these management strategies” mentioned? Clarification has to be given.

 4) The extended arrangement of Tables at the methodology followed, it has to be accompanied by explanatory text of functionality, based on (but not merely reiterating) the data referred. Each one Table has to be accompanied by one or two explanatory paragraphs, respectively.

 5) The “Conclusions” are actually referred to the methods applied and the discussion upon results, but actually the concluding remarks are not strongly supported, no fully verified, by the preceding analysis. Therefore, it is recommended authors to reorganize this text to a new “Discussion” section, which is missing, and to formulate a concise, no more than two paragraphs text, in which the main limitations, shortcomings, challenging issues, and future research orientations are driven, into a wider socio-economic and cultural applicability of the remarks to other (than Korean) developing economies. The “Conclusions” section is no cited, whereas the formulated “Discussion” section is a cross-cited text and, to this end, authors are recommended to proceed in: literature refreshment and citing those extra papers to support their argumentation developed at this “Discussion” section. Moreover, at this “Discussion” section, the diagrammatic/schematic/graphical representation of the main independent- and depended- variables can be shown/developed, accordingly.

Round 2

Reviewer 2 Report

Accept.

Author Response

Thank you for your kind review of the paper.

Reviewer 3 Report

At this revised manuscript the authors proceeded in a careful and meticulous revision of their research work, while considering the reviewers’ comments in a constructive manner. In this respect, the manuscript is well organized and of insightful remarks upon the trading and economic status of the Korean economy. The revised manuscript can be published at the “Journal of Open Innovation: Technology, Market, and Complexity” as is. Just at the proof-editing process:

1) the section of “6. Discussion” to be relocated prior to the “5. Conclusions” section. Therefore, the row of sections should be as: …..section 5: Discussion…., then, section 6: Conclusions…….

2) The citation of Malesias et al [40] at line 446 to be changed to the correct: Malesios et al. [40].

Author Response

Thank you for your kind review of the paper. The modifications are as follows.

1) the section of “6. Discussion” to be relocated prior to the “5. Conclusions” section. Therefore, the row of sections should be as: …..section 5: Discussion…., then, section 6: Conclusions…….

--> We changed the order as we informed you.

2) The citation of Malesias et al [40] at line 446 to be changed to the correct: Malesios et al. [40].

--> I have corrected the Malesios et al.[40].